# Allostery Inhibition of BACE1 by Psychotic and Meroterpenoid Drugs in Alzheimer’s Disease Therapy

**DOI:** 10.3390/molecules27144372

**Published:** 2022-07-08

**Authors:** Samuel C. Ugbaja, Isiaka A. Lawal, Bahijjahtu H. Abubakar, Aganze G. Mushebenge, Monsurat M. Lawal, Hezekiel M. Kumalo

**Affiliations:** 1Discipline of Medical Biochemistry, School of Laboratory Medicine and Medical Science, University of KwaZulu-Natal, Durban 4001, South Africa; aganzedar@gmail.com (A.G.M.); lawalmonsurat635@gmail.com (M.M.L.); 2Chemistry Department, Faculty of Applied and Computer Science, Vanderbijlpark Campus, Vaal University of Technology, Vanderbijlpark 1900, South Africa; lawalishaq000123@yahoo.com; 3The Renewable Energy Programme, Federal Ministry of Environment, Aguiyi Ironsi St, Maitama, Abuja 904101, Nigeria; bahijjah@yahoo.com

**Keywords:** Alzheimer’s disease, BACE1, multisite targeting, allosteric inhibitor, molecular docking, molecular dynamics (MD) simulations

## Abstract

In over a century since its discovery, Alzheimer’s disease (AD) has continued to be a global health concern due to its incurable nature and overwhelming increase among older people. In this paper, we give an overview of the efforts of researchers towards identifying potent BACE1 exosite-binding antibodies and allosteric inhibitors. Herein, we apply computer-aided drug design (CADD) methods to unravel the interactions of some proposed psychotic and meroterpenoid BACE1 allosteric site inhibitors. This study is aimed at validating the allosteric potentials of these selected compounds targeted at BACE1 inhibition. Molecular docking, molecular dynamic (MD) simulations, and post-MD analyses are carried out on these selected compounds, which have been experimentally proven to exhibit allosteric inhibition on BACE1. The SwissDock software enabled us to identify more than five druggable pockets on the BACE1 structural surface using docking. Besides the active site region, a melatonin derivative (compound **1**) previously proposed as a BACE1 allostery inhibitor showed appreciable stability at eight different subsites on BACE1. Refinement with molecular dynamic (MD) simulations shows that the identified non-catalytic sites are potential allostery sites for compound **1**. The allostery and binding mechanism of the selected potent inhibitors show that the smaller the molecule, the easier the attachment to several enzyme regions. This finding hereby establishes that most of these selected compounds failed to exhibit strong allosteric binding with BACE1 except for compound **1**. We hereby suggest that further studies and additional identification/validation of other BACE1 allosteric compounds be done. Furthermore, this additional allosteric site investigation will help in reducing the associated challenges with designing BACE1 inhibitors while exploring the opportunities in the design of allosteric BACE1 inhibitors.

## 1. Introduction

Alzheimer’s disease (AD) is a multifactorial neurodegenerative disorder discovered in November 1906 [1,2]. The extracellular β-amyloids (Aβ) plaques and intracellular neurofibrillary tangles (NFTs) are cardinal AD pathological features [1,3,4]. β-amyloids aggregate when either β-secretase or γ-secretase cleaves the amyloid precursor protein (APP). β-amyloids accumulation can also occur when both β and γ secretase proteins concurrently (Figure 1) cleave APP. At the N-terminal of the APP, the β-secretase cleavage results in more harmful soluble β-amyloid precursor protein (sAPPβ) and C99 amino acid fragments [1]. The amino acids produced by β-secretase mainly consist of Aβ-40 [1,5], which are often cleared by a combination of lysosomal and protease processes or conjugates and exhibit harmful effects. Subsequently, the cleavage by γ-secretase at the C-terminal results in peptides of the length 30–43 Å [1,5]. The cleavage by γ-secretase is regarded as non-amyloidogenic (Figure 1) because it produces nonreactive P3 peptide (in the middle segment) in addition to intracellular carboxy-terminal fragment (CTF) [6,7]. Further cleaving of APP at the extracellular domain with β-secretase or α-secretase at the distal region results in the amyloidogenic pathway. This cleavage produces β-amyloid fragment plaques in the middle segment with a soluble sAPPβ N-terminal [6,7].

Alzheimer’s disease facts and figures (2021) showed that about 6.2 million people in the United States of America are living with AD-related dementia and this incidence might increase to about 13 million by the year 2050 [1,8,9]. Globally, approximately 600 billion American dollars are required to care for a 35 million population living with AD-related dementia per annum, and this amount is reportedly 1% of global Gross Domestic Product (GDP) [1,10].

**Figure 1 molecules-27-04372-f001:**
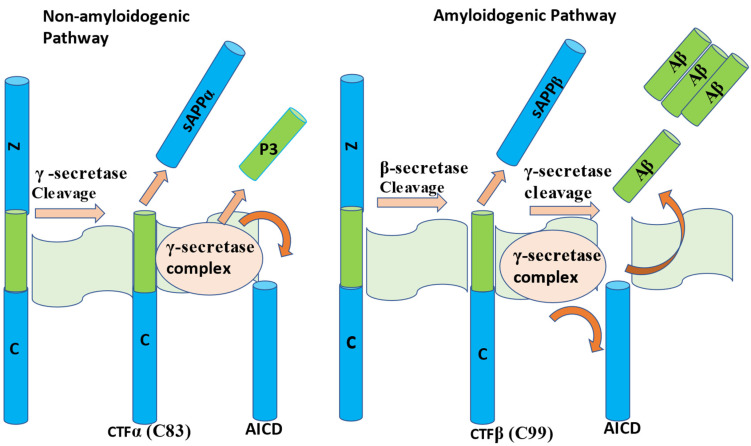
Illustration of APP amyloidogenic and non-amyloidogenic pathways [1,11]. The cleavage mechanism by γ-secretase is non-amyloidogenic, while β-secretase or α-secretase cleavage results in the amyloidogenic pathway. CTFα and CTFβ are alpha and beta carboxy-terminal fragments, AICD represents the amyloid precursor protein intracellular domain.

The early symptoms of AD include disruption of daily life activities due to memory loss; difficulty in understanding visual images and completing known tasks; challenges with identification, speaking, and writing new words; disorganization; and constant mood swings [12,13]. Available AD treatments have focused on decreasing the level of Aβ aggregation and the accumulation of hyperphosphorylated tau. An alternative approach is using drugs to reverse the symptoms, prevent tissue/cell damage, reduce fats/swollenness, inoculate, and for hormonal treatment [14]. Currently, some tau-related therapies are under clinical trial. For an efficient AD treatment approach, early treatment assessment strategies and diagnostic biochemical markers are crucial [15]. The current research on AD treatment is three-fold. This includes identifying populations with higher vulnerability (persons > 60 years) and helping them with the preventions intended to modify the risk factors. Furthermore, taking advantage of the modern neuroimaging and cerebrospinal fluid examinations and subsequent data analysis in early diagnosis of the persons at the preclinical stage. And finally, the identification of disease-modifying compounds to inhibit Aβ and the NFTs accumulation [16].

Since its discovery in 1999, BACE1 has had a primary focus on the reduction of β-amyloid plaques in the brain. Although there is no approved BACE1 inhibitor [17], evidence of its inhibition towards AD management remains feasible. Its inhibition is one of the therapies targeted toward AD management [18,19]. Researchers have made relentless efforts to design potential BACE1 inhibitors with desirable properties in the reduction of Aβ accumulation in the brain. Six potential BACE1 drugs such as Atabecestat, Umibecestat, LY3202626, Elenbecestat, Lanabecestat, and Verubecestat, showed initial benefits in reducing the amyloid-beta plaques responsible for AD-related symptoms [18,19,20]. Several complications from these BACE1 drugs at phase II/III clinical trials informed discontinuation [21,22]. There is a need to understand why these BACE1 drugs failed. Good knowledge of the inhibiting potentials and properties of BACE1 inhibitors serves as a prerequisite for the development of molecules with high selectivity and specificity [18,19,20]. Also, more attention on BACE1 non-active sites inhibition might be a promising approach in targeting them in AD management.

BACE1 is a family of the aspartate protease with a characteristic catalytic dyad (Asp32/228) at its active site, required for substrate cleavage [23,24]. The β-hairpin loop (flap), through its dynamic conformations, controls the movement of the substrate into the active site. The flap residues are 67–75 besides the binding subsites S1, S2, S3, S4, S1′, S2′, S3′, and S4′. Residues Leu30, Phe108, Ile110, Ile118, and Trp115, within S1 and S3 subsites, make up the hydrophobic region, while S2 and S4 hydrophilic region comprise residues Lys9, Ser10, Thr72, Gln73, Thr231, Thr232, Arg235, Arg307, and Lys321. Other hydrophilic sites include the S3′ and S4′ with residues Pro70, Thr72, Glu125, Arg128, Arg195, and Trp197. The S2′, which is amphipathic (both hydrophobic and hydrophilic) and located closer to S4′, contains residues Ser35, Val69, Ile126, and Tyr198. The catalytic dyad is at the center of S1′ (with residues Ile226 and Val332) subsite [24,25]. Figure 2 shows the summary BACE1 binding pockets, substrate cleavage, and anti-BACE1 binding to the non-catalytic region.

Allosteric inhibition is a form of noncompetitive inhibition that occurs at another protein site different from the active site [26]. In allosteric inhibition, the inhibitor binds to a distant site away from the active site, rendering the active site unfit for the substrate to bind [26,27]. We lately reviewed different studies from researchers on BACE1 allosteric sites inhibition and exosite-binding antibody development [20]. The survey facilitates the documentation of some inhibitors with potential propensities to the BACE1 allosteric sites (Figure 3). Most of these compounds identified using experimental approaches are yet to be explored at the molecular level to evaluate if they indeed interact at the allosteric regions of BACE1. However, some phytocompounds, including 3,5,7,3′,4′-pentamethoxyflavone (PMF) [28], nor-rubrofusarin 6-*O*-β-d-glucoside [29], loganin [30], and gamma-linolenic acid [31] have been proposed through docking and kinetics study as allosteric inhibitors. Besides, Rombouts et al. [32] applied a fragment-based virtual screening approach to identify BACE1 inhibitors, which bind at the other subsites without interacting with Asp32 and Asp228. Integrated analysis, including nuclear magnetic resonance (NMR), fluorescence resonance energy transfer (FRET) assay, and ThermoFluor (TF), produced six hits [32]. Refinement and analysis showed that four of these compounds [32] competitively bind with **OM99-2** [33]. X-ray atomistic interaction revealed that one of these hits occupied the S1 and S3 subsites without interacting with the catalytic Asp32 and Asp228 (Figure 3). Compound **12** with protein data bank (PDB) [34] code 5MXD [32] showed an IC_50_ value of 0.5 mM, which is significant. Although the interaction pose occurred close to the BACE1 active site region, we assume this mechanism as a potential allosteric inhibition.

Herein, we apply molecular modeling methods, including molecular docking and molecular dynamics (MD) simulations to unravel the interaction of some potential allosteric inhibitors with BACE1. Readers may be puzzled by the uniqueness of the present work since there exist investigations on BACE1 allosterism. Indeed, there are propositions on BACE1 allosteric or secondary sites inhibition, there is still a knowledge gap at the molecular level using computational methods, like MD simulations. Besides, Pietro et al.’s (2017) study of conformational ensemble and binding mode analysis of some multisite inhibitors using MD and docking method [36], studies that explore multisite targeting of BACE1 towards drug design for AD are still scarce. In this study, we selected compounds (Figure 4) proposed by different authors [28,29,30,31,37,38,39,40,41] as potent BACE1 allosteric inhibitors for some computational experiments. Docking analysis revealed that these compounds bind at allosteric sites but are highly stable at the active site. Our observation aligns with the kinetic experiments and suggestions available in the literature [28,29,30,31,37,38,39,40,41] for the various compounds. Only compound **1** (a melatonin derivative) with 88% inhibitory activity at 5 μM BACE1 concentration [37] showed appreciable stability at six different allostery pockets. We further refined the stability of the representative molecule (compound **1**) at the BACE1 binding region to establish if these sites are transient or long-lived using several metrics. Understanding how potent small inhibitors modulate and inhibit BACE1 at the molecular level would enable us to manipulate synthetic enzymes or design drugs for AD treatment.

## 2. Materials and Methods

### 2.1. System Preparation

BACE1 X-ray 3D structures are available as several complexes in the PDB repositories. We selected 6PZ4 [42] to maintain consistency with previous studies [1,43]. Investigations have shown favorable outcomes with BACE1 mono-protonation of Asp32 [1,44], hence, we protonated Asp32 using PROPKA [45] at pH 7. Protein structure refinement entails using the Maestro protein preparation wizard package [46] to add hydrogens, assign bond orders, remove water and non-ligand molecules. We added missing residues in MODELLER 9.19 [47] implemented in the UCSF Chimera [48]. The ligands were prepared for docking in GaussView 6.0.16 [49], optimized in Avogadro [50], and saved in Mol2 format for the docking study.

### 2.2. Docking

Docking is a molecular modeling method that involves predicting the preferred orientation of one molecule when bound to another molecule [51]. It requires computational software compatible to dock small compounds into a macromolecule including protein structures [52]. The conformations of the docked ligand-enzyme complex prediction involve assessing the different poses of the ligand within the receptor’s binding site. The scoring of the various poses enables predicting the molecular mechanisms, the binding free energy of the complex, and nonbonded interaction properties [53]. Over the years, the combination of docking and further binding free energy predictions remained useful in designing protease inhibitors such as HIV and BACE1 proteases [54,55].

We used SwissDock online docking software [56,57] to search for the regions where the selected compounds bind to the BACE1. SwissDock automatically prepares the ligand and enzyme structures before docking using a webserver. The implemented docking algorithm is the CHARMM force field [58] to offer an accurate docking approach. Therefore, ligands and the protein files (in Mol2 and PDB) automatically convert to CHARMM format after upload. Although prediction of ligand poses are premised on experimentally identified binding sites, the SwissDock is advantageous because it enables flexible docking [56,57]. This unconstrained docking facilitates several ligand conformational orientations within the enzyme. The automated process also reduces human errors by using the web interface in generating alternative input files and parameters while interpreting the docking outcome [56]. SwissDock webserver is incorporated with the Apache web server, PHP (opensource technology) [59], which has dual Xeon E440 2.83 GHz at 1.7 Å and 16 GB. The docking procedures include minimizing at 100 steps steepest descent to remove any clashes inherent from adjusting the protein structure. Also, it uses a 5 kcal/mol.Å^−2^ constraint to restrict nonphysical movements of heavy atoms during minimization.

### 2.3. Molecular Dynamic Simulations

MD MD simulation is an approach in computational modeling to explore the conformational dynamics of molecules [60]. Molecular simulation provides a similitude of real-time possibilities to predict protein or molecule behavior with or without interacting with other molecules [61]. MD simulation is crucial in drug design; it facilitates binding and catalytic mechanism prediction after identifying a molecule with a plausible propensity for a target [62,63]. Many researchers have indicated MD as crucial to refine, validate, and improve docking evaluation [61,64]. There are several commercial, non-commercial, and web-based MD simulation programs for molecular modeling studies [63] in which the AMBER suite [65] is a prominent one.

We performed MD simulations on the AMBER 18 program package integrated with graphics processing units (GPU) [66] using the particle mesh Ewald method (PMEMD) [67] package with the Sander module. The long-range electrostatic interactions cut-off is at 12 Å. The simulation pre-step involves generating a partial atomic charge for the ligand using the General Amber Force Field (GAFF) [68] of the ANTECHAMBER module. The GAFF is a simple harmonic function developed as a complete and suitable force field for rational drug design [69]. Further procedures are topology and coordinate preparation, enzyme-ligand coupling, solvation, and neutralization. Explicit solvation was done with the Leap module of the AMBER 18 package in a TIP3P orthorhombic water box at 10 Å to any edge. The pre-MD production steps are partial minimization, total minimization, heating, and equilibration. 

The partial minimization was at 10,000 steepest descent steps with 10 kcal/mol.Å^−2^ harmonic restraint on all heavy atoms to relax the system and remove potential atom clashes. We also run a full minimization with another steepest and conjugate gradient descents at 5000 steps each without constraints. Systems heating was from 0 K to 300 K for 300 ps using Langevin dynamics, 1 ps collision frequency, and a 5 kcal/mol.Å^−2^ applied harmonic restraints at a constant volume. We subsequently equilibrated for 500 ps at 300 K under constant pressure and temperature (NPT) ensemble. The final step is MD simulation for 120 ns at 2 fs time step without any restraints at 300 K and 1 atm in the NPT ensemble switching on the Langevin temperature scaling [70] and Berendsen barostat [71] algorithm for temperature and pressure, respectively. The molecular dynamic simulation also involves applying the SHAKE algorithm [72] to constrain hydrogen atoms.

### 2.4. Post-Simulation Analysis

We measured system stability through root-mean-square deviation (RMSD) calculation. The RMSD trajectory of the protein backbone alpha carbon (Cα) generated with the CPPTRAJ [73] module uses Equation (1) for its estimation. The standard deviation of the interatomic distance between Cα backbone atoms of two amino acids v and w at n points in Equation (1) represents v_i_ as Cα coordinates in v at the time i, and w_i_ is the coordinates of Cα atom in w at the time i.
(1)$$\mathrm{RMSD}\left(v,w\right)=\sqrt{\frac{1}{n}\overset{n}{\underset{i=1}{\sum }}||v_{i}-{\;w}_{i}}|{|}^{2}\;$$

The radius of gyration (RoG) is the moment inertia of atoms from their center of mass. RoG is often applicable in quantifying the molecular rigidity of a system [74]. The RoG (Equation (2)) is the square root of the inertia moment of atoms, where n is the number of atoms, r_i_ represents the atomic position, and r_m_ signifies the mean position of all atoms.
(2)$$\mathrm{RoG}=\sqrt{\frac{1}{n}\overset{i=0}{\underset{n}{\sum }}(r_{i}-{\;r}_{m}}{)}^{2}\;$$

We estimate the root-mean-square fluctuation (RMSF) to predict the conformational changes on a per-residue basis. Equation (3) shows the RMSF equation whereby x_i(j)_ represents the i-th Cα atom position in the j-th model structure, and (x_i_) denotes the averaged location of the i-th Cα backbone atom in all models.
(3)$$\mathrm{RMSF}=\sqrt{\frac{1}{n}\overset{n}{\underset{j}{\sum }}|x_{i\left(j\right)}-(x_{i}}){|}^{2}\;$$

Further analyses include secondary structure prediction with the definition secondary structure of protein (DSSP) approach of Kabsch and Sander [75] implemented in the CPPTRAJ program. The DSSP approach involves analyzing the most likely secondary structure assignment through the 3D structure of a protein. It entails interpreting atom position in a protein and calculating the hydrogen bond (HB) energy between all atoms. DSSP algorithm ignores any hydrogen available in the input structure then computes the significant hydrogen positions after placing them at 1.0 Å from the backbone nitrogen in the reverse direction from the backbone carbonyl bond [75]. The assignment completes by using the top two HBs in N and C=O to predict the most likely secondary structure class for each residue in the protein.

## 3. Results and Discussion

### 3.1. Allosteric Sites and Prediction

Blind molecular docking with SwissDock generated different binding poses at various sites on the BACE1 protein structure. All the compounds (Figure 4) show favorable interaction with BACE1 widely around the active site region and sparsely at other domains (Figure 5). This observation indicates that these compounds are selective for the BACE1 active site over other subsites. We notice a significant overlay of **LY2811376** poses on the BACE1 flap tip as the only identified allostery site. The observation is consistent with previous reports [38,39] on **LY2811376** modulatory and inhibitory potentials. All the predicted sites are feasible subsites as suggested in a survey of ligand-binding modes of co-crystallized BACE1 structures [76] and molecular modeling study [36].

Zoomed-in profile of all compounds’ interactions at the BACE1 active site is available in the supporting information (SI) captioned as Appendix A. Gamma-linolenic and sargahydroquinoic acids with the most favored binding affinity (Table 1) at BACE1 active site show two distinct hydrogen bond (HB) interactions (Appendix A) via Thr33/Gly34 and Gly34/Gln73 (Table 2), respectively. Electrostatic interaction from hydrogen bonding is imperative for inhibitor stability at the active site of an enzyme. The electrostatic effect represents an approach to predict catalysis [77]. Compound **1** shows an interaction at eight different subsites on the BACE1 scaffold (Figure 6), thereby indicating the molecule as a potent allostery modulator and multitarget directed ligand (MTDL). The compound is relatively sizeable compared to others (Figure 4) with unique functional groups, including amide. These physicochemical properties make compound **1** flexible with the propensity to attach at several regions on the BACE1 scaffold through different molecular interactions, including electrostatic and van der Waals (Figure 6).

Table 1 shows the full fitness energy and the predicted binding scores for all the potent molecules’ interactions at the active site. The binding poses scored using their full fitness and clustered show interaction energy values within −7.51 and −8.64 kcal/mol. Although with the lowest binding score, compound **1** has the most favored full fitness of its moieties in the BACE1 active site. Its binding energy to other binding regions within the enzyme ranges from −6.12 to 7.10 kcal/mol with favorable fitness scores (Figure 6). The interaction of Thr232 and flap region residues buttress previous molecular docking prediction of compound **1** binding pose [37]. To give a clearer picture of the stability of compound **1** interaction to BACE1 at the different binding subsites, we simulate each complex (Figure 6) using the amber force field. The molecular docking approach enables us to unravel druggable regions for potential multisite targeting in BACE1 towards AD treatment.

### 3.2. Protein Structural Changes

We assess changes to the BACE1 backbone structure when bound to compound **1** using RMSD calculation of the protein Cα atoms. In molecular modeling, RMSD scoring enables conformational dynamics and system stability prediction. Figure 7 depicts the RMSD plots per time in which allosteric sites **2** and **3 **(Figure 6) are unavailable because their MD simulation failed. The data indicate the relative stability of the protein’s primary structure over the 120 nanoseconds production run. Each complex shows appreciable overlap with relative stability around 90–120 ns, indicating that all the protein structures converged without large-scale conformational transitions. RMS deviation higher than 3.5 Å is a potential indication of significant BACE1 conformational switch in domains such as flap opening and closing [43].

Compound **1** binds to the active site with an average RMSD of 2.162 ± 0.206 Å, while allosteric sites **1**, **4**–**8** show mean RMSD scores within 1.766 and 2.072 Å (Table 3). This result denotes that compound **1** binding at the active site increases the protein backbone dynamics compared to the six potential allosteric sites. Note that the minimum RMSD value is 0 Å, and allostery site **4** shows the lowest outcome in all the parameters (Table 3), while the free BACE1 structure has an average RMSD value of 3.75 ± 0.44 Å [43].

To estimate the effect of compound **1** interaction on BACE1 intrinsic arrangement during the simulation, we estimate the protein secondary structure using the DSSP [75] method. Figure 8 shows that the ligand binding at the various spaces induces small changes to the protein structure. For instance, the binding of compound **1** at allosteric site **1** diminishes the alpha arrangement to anti around residues 335–338. BACE1 has a few highly folded alpha helixes (pink color, Figure 5). Alteration of the helix to anti (Figure 8) might impact the protein’s structural stability significantly. The DSSP map of allosteric sites **4**–**8** (Appendix A) is like the active site (Figure 8).

### 3.3. Allostery Dynamics

We use the RMSF metric to probe how compound **1** dynamically perturbs the motion of the entire protein structure. The computation includes all atomic flexibility (both backbone and sidechain atoms) to identify the effect of the studied molecule at the various regions on BACE1 per residue. RMSF is a crucial concept to evaluate protein dynamics and individual residue flexibility [78]. The RMSF projections for compound **1** binding (Figure 9) are fascinating, with a high spectrum separation between the active site and others (potential allosteric sites). This outcome signifies that compound **1** binding to allostery sites decreases BACE1 flexibility, whereas residue mobility is high when bound to the active site. The fluctuation pattern and projection for active site binding (Figure 9) are akin to our previous simulation of apo BACE1 RMSF [43], denoting that compound **1** binding at the active region is most likely inconsequential on the protein mobility, thus supporting Panyatip et al. [37] in silico predictions. Eighteen residues show very high fluctuations (>20 Å) with the highest RMSF of 23.422 Å (see maximum value in Table 4) in Ser58 for the active site system.

We propose that the allostery mechanism of compound **1** on BACE1 is both modulatory and inhibitory. The analysis reflects that the melatonin derivative can allosterically restrict BACE1 motility. It acts as an inhibitor at the potential allosteric sites and as a modulator at the active site. This phenomenon indicates that compound **1** is a potential MTDL targeting multiple subsites on BACE1. The mechanism would distort BACE1 structural arrangement and availability for natural substrate binding, lowering its level in AD conditions. Irrespective of the targeted subsite, compound **1** exerts relatively low flexibility across all the 385 residues. The average RMSF values are within 1.095 and 1.348 Å (Table 4), with allosteric site **7** showing slightly higher projections (above 5 Å) at residues 164–167.

### 3.4. Rigid Body Motion

The mass-weighted radius of gyration (RoG) is a moment of atom inertia from their center of mass [79]. The calculation setup for RoG analysis includes all the atoms to notice the effect of allostery binding on the BACE1 surface. Figure 10 shows the RoG evolution per time for each model system. Besides sites **7** and **8** with average RoG of 21.5 and 21.4 Å, the approximate average RoG value is 21.2 Å (Table 5). These values are close, denoting that the atoms in the BACE1 structure have similar moment inertia in the various models. However, allosteric site **7** facilitates slightly higher RoG (Figure 10) like its RMSF (Figure 9). The data signifies less compactness in the allosteric site **7** BACE1 model.

Also, allostery site **7** shows the highest data in all the parameters, while site **5** shows the least mean RoG score (Table 5). The RoG metric enables us to predict how BACE1 moves as a rigid body when complexed with the melatonin derivative at the active region and allosteric sites. The outcome shows that compound **1** exerts compactness on the BACE1, restricting protein motions during the simulation. The similarity of the RoG projections and the same mean values depicts compound **1** as a potential active site and allosteric inhibitor. This outcome denotes that the computationally predicted sites are potential target sites for drug binding in BACE1.

## 4. Conclusions

Allosteric inhibition is a type of noncompetitive inhibition occurring at another protein site different from the active site. In this mechanism, the inhibitor binds to a site(s) other than the active site, thereby rendering the active site unfit for the substrate to bind. In allosteric inhibition, it is a case in which the moiety gets to the enzyme or protein first, which blocks the other from binding. As a sequel to our previous review study on some identified allosteric inhibitors and exosite-binding antibodies of BACE1 over the last 8 years (2013–2020), we, hereby, in this study applied CADD methods to further validate the claims as contained in the reviewed study. We selected six compounds that were reported to bind on sites different from the BACE1 active site. Some of these compounds are classified as meroterpenoids (Sargahydroquinoic acid and Gamma-Linolenic acid) and psychotic drugs (Anisoperidone). Good knowledge of the inhibiting potentials and properties of BACE1 inhibitors also serves as a prerequisite for the development of molecules with high selectivity and specificity. We show in this study that besides the active site, BACE1 targeting at other subsites is plausible towards AD therapy. From the results of the molecular docking, molecular dynamic (MD) simulations, and subsequent post-MD analyses, the identified regions correspond with some predicted subsites and the small molecules (like compound **1**) bind through several other BACE1 residues. The allostery and binding mechanisms of the selected potent inhibitors show that the smaller the molecule, the easier the attachment to several enzyme regions. This finding hereby establishes that most of these selected compounds failed to exhibit strong allosteric binding with BACE1 except for compound **1**. We hereby suggest further studies and additional identification/validation of other BACE1 allosteric compounds be done. Furthermore, this additional allosteric site investigation will help in reducing the associated challenges with designing BACE1 inhibitors, while exploring the opportunities in the design of allosteric BACE1 inhibitors. We further suggest that diverting attention from the conventional active site through a detailed computer-aided drug design approach would assist in designing a potential BACE1 inhibitor that might attain the approval stage for AD management.

## Figures and Tables

**Figure 2 molecules-27-04372-f002:**
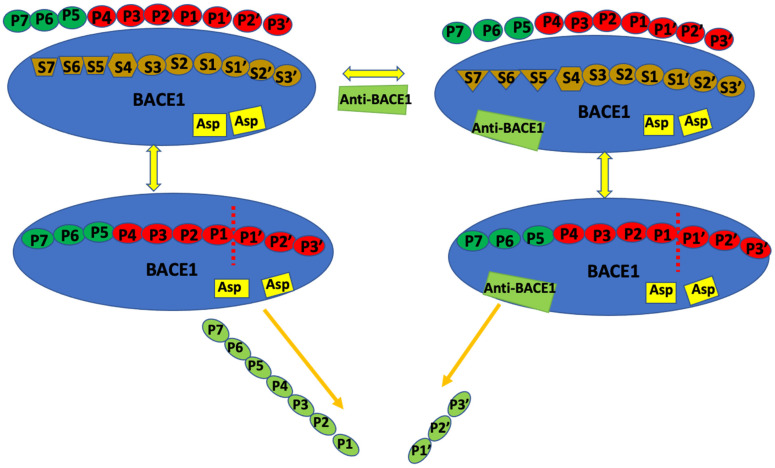
Illustration of BACE1 active site (primary binding site), binding subsites (secondary sites), and anti-BACE1 attachment [20]. The characteristic catalytic Asp32/228 dyad hydrolyzes the substrate or reversible inhibitor; other non-catalytic residues are involved if an inhibitor binds at secondary sites.

**Figure 3 molecules-27-04372-f003:**
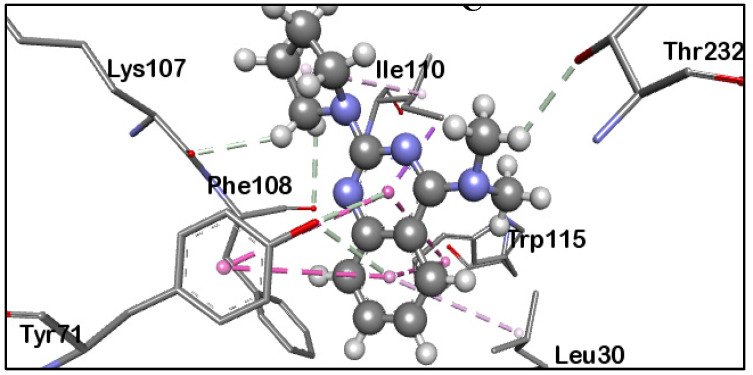
Zoomed-in 3D snapshot of BACE1 crystal structure complexed with compound **12** (PDB code: 5MXD) devoid of Asp32/228 interaction [32]. The compound binds at the flap region of BACE1 and we generated this image with the Discovery Studio R2017 [35].

**Figure 4 molecules-27-04372-f004:**
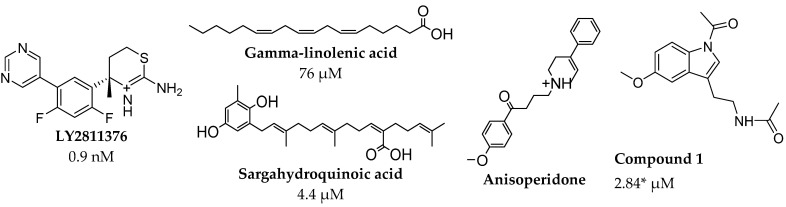
The selected potential allosteric inhibitors of BACE1 with reported IC_50_ values (no data for anisoperidone and * indicates extrapolated from 88% inhibition of BACE1 at 5 μM). Besides **LY2811376**, all the molecules are naturally occurring compounds or their derivatives. **LY2811376** 0.9 nM [38,39]. Gamma-linolenic acid 76 μM [31]. Sargahydroquinoic acid 4.4 μM [40]. Anisoperidone [41]. Compound **1** 2.84* μM [37].

**Figure 5 molecules-27-04372-f005:**
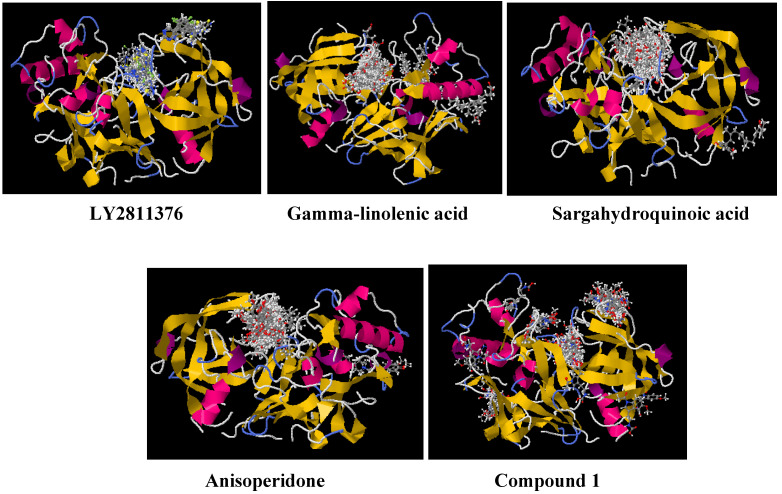
Binding modes of the allostery inhibitors on BACE1 through docking. **LY2811376** converges mainly at the active site and partially on top of the flap and gamma-linolenic acid binds favorably at the primary site and two other secondary sites as predicted previously [31]. Sargahydroquinoic acid and anisoperidone interact at the active site and extend to the flap tip, while compound **1** binds at several regions.

**Figure 6 molecules-27-04372-f006:**
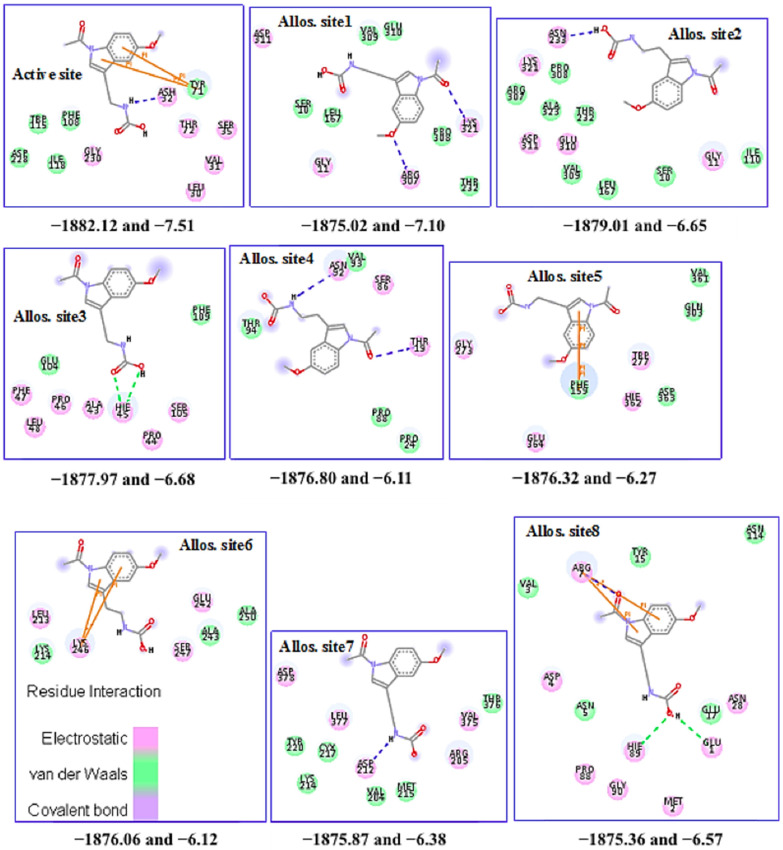
The interaction profile of compound **1** for its various poses on the BACE1 scaffold. The first and second values below each pose are full fitness and binding energy in kcal/mol, respectively. Allos. site signifies allosteric site, the dashed green and blue lines represent classical hydrogen bond and bridging water HB, respectively.

**Figure 7 molecules-27-04372-f007:**
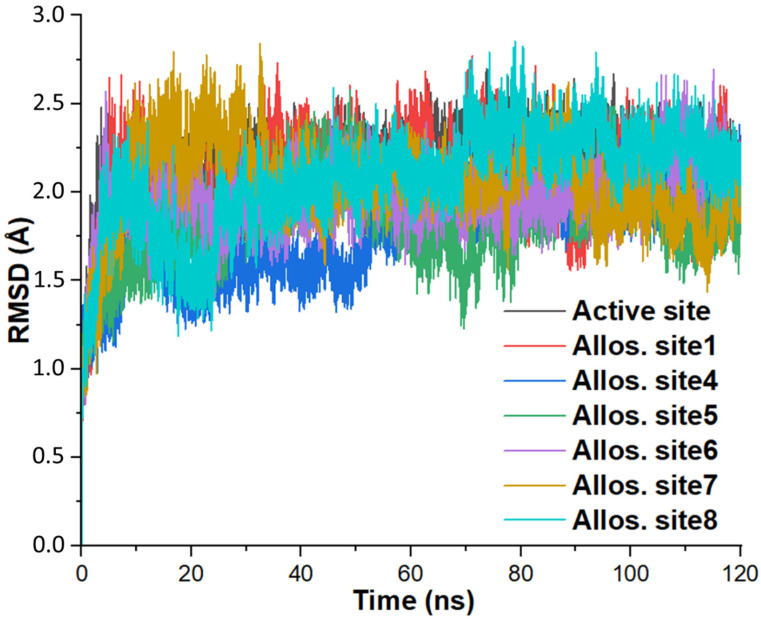
Evolution of the alpha carbon backbone atom root-mean-square deviation (RMSD/angstrom) over 120 ns molecular simulations for compound **1** binding to various regions on BACE1 structure. Allos. site represents the allosteric site and the definition for each site including the interaction residues is available in Figure 6.

**Figure 8 molecules-27-04372-f008:**
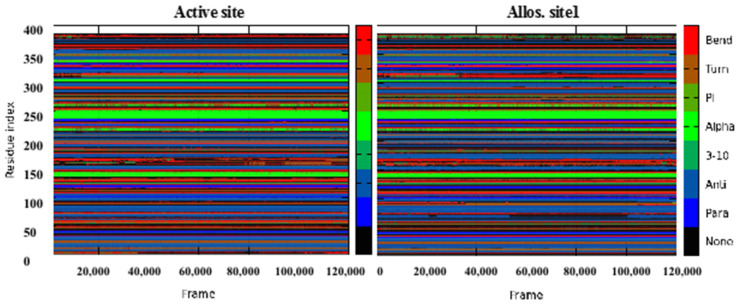
Protein secondary structure prediction per 385 residues over time 120 ns (120,000 frames) for compound **1** binding to BACE1 active site and allosteric site **1**.

**Figure 9 molecules-27-04372-f009:**
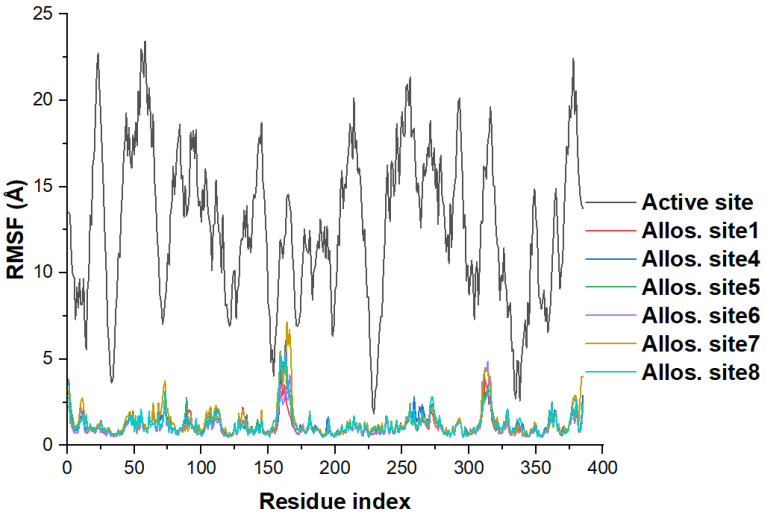
Per residue (a total of 385) fluctuation through all-atom root-mean-square fluctuation (RMSF/angstrom) scoring for compound **1** binding to several regions on BACE1. Allos. site represents the allosteric site and the definition for each site including the interacting residues is available in Figure 6.

**Figure 10 molecules-27-04372-f010:**
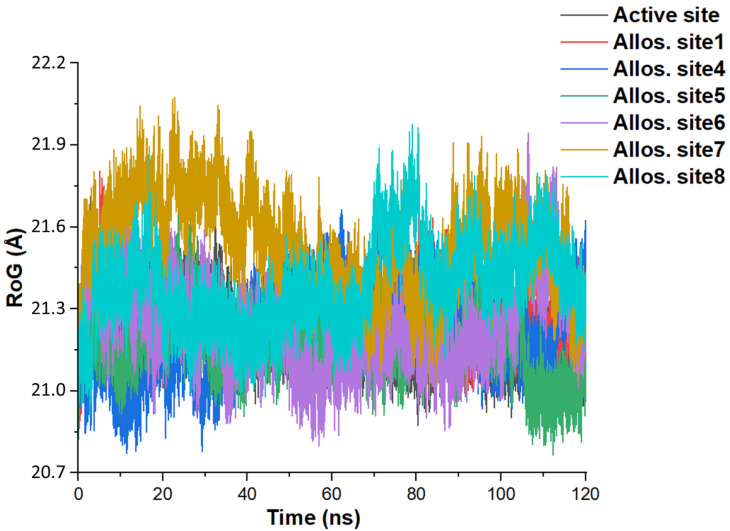
Time (in nanoseconds) evolution of the mass-weighted RoG—radius of gyration (in angstrom) induced by coupling compound **1** to the various regions on BACE1 structure. Allos. site represents the allosteric site and the definition for each site including the interacting residues is available in Figure 6.

**Table 1 molecules-27-04372-t001:** Full fitness and binding energy (in kcal/mol) predicted from docking of potent allostery molecules to BACE1 active site. The co-crystalized ligand (AM6494) of 6PZ4 binds preferentially at the catalytic active site of BACE1 as proposed [42] and validates the accuracy of the docking method.

	AM6494 (Internal Ligand)	LY2811376	Gamma-Linolenic Acid	Sargahydroquinoic Acid	Anisoperidone	Compound 1
Full fitness	−1610.66	−1868.32	−1868.37	−1849.84	−1814.99	−1882.12
Binding energy	−8.47	−7.80	−7.97	−8.64	−7.91	−7.51

**Table 2 molecules-27-04372-t002:** Predominant interactions of some of the docked compounds including their binding energies.

Compound	Class	Binding Energy in kcal/mol	Residues Forming Hydrogen Bond	Residues Forming Electrostatic Interaction
Sargahydroquinoic acid	Meroterpenoids	−8.64	Gln73, Gly34	Gly74, Gly73
Gamma-linolenic acid	Meroterpenoids	−7.97	Gly34, Thr33	Val31, Ash32, Thr33, Gly34, Ser35
Ly2811376	Antipsychotic drug	−7.80	Ash32	Leu30, Val31, Ash32, Gly34, Thr72, Gln73, Gly74, Lys107, Phe108, Leu119, Asp228
Anisoperidone	Psychotic drug	−7.91	Ser35	Ash32, Ser35, Gly230

**Table 3 molecules-27-04372-t003:** Analysis of the RMSD (in Å) scores over the 120 ns simulation time. Std. represents standard, and zero is the minimum RMSD value in all cases. Allos. site equals allosteric site.

Structure	Mean	Std. Deviation	Sum	Median	Maximum
Active site	2.1621	0.2055	259,457.3803	2.2042	2.6966
Allos. site1	2.0978	0.2361	251,733.1464	2.1311	2.7724
Allos. site4	1.7662	0.2415	211,943.7953	1.8211	2.3821
Allos. site5	1.8605	0.2358	223,259.2948	1.8680	2.5829
Allos. site6	1.9718	0.1835	236,610.3866	1.9719	2.6936
Allos. site7	2.0477	0.2511	245,728.2049	2.0577	2.8403
Allos. site8	2.0715	0.2863	248,582.0876	2.1185	2.8550

**Table 4 molecules-27-04372-t004:** Statistics of the RMSF (in Å) scores per residue. Std. represents standard and Allos. site represents the allosteric site and the definition for each site including the interaction residues is available in Figure 6.

Structure	**Mean**	**Std. Deviation**	**Sum**	**Minimum**	**Median**	**Maximum**
Active site	12.7287	4.4411	4900.5407	1.8308	12.7957	23.4217
Allos. site1	1.1061	0.5945	425.8476	0.4607	0.9276	4.6406
Allos. site4	1.2339	0.6990	475.0518	0.4862	1.0253	6.0701
Allos. site5	1.1837	0.7235	455.7218	0.4885	0.9607	5.4793
Allos. site6	1.0945	0.6632	421.3987	0.4608	0.8835	4.8497
Allos. site7	1.3478	0.9132	518.9077	0.4959	1.0523	7.1664
Allos. site8	1.2023	0.6809	462.8932	0.4824	1.0027	5.1997

**Table 5 molecules-27-04372-t005:** Data analysis of the RoG (in Å) scores over the simulation time (120 ns). Std. represents standard, and Allos. site indicates allosteric site.

Structure	**Mean**	**Std. Deviation**	**Sum**	**Minimum**	**Median**	**Maximum**
Active site	21.2192	0.1077	2.5463 × 10^6^ 2.5463 × 10^6^	20.8742	21.2056	21.7118
Allos. site1	21.2645	0.1095	2.5517 × 10^6^	20.8634	21.2516	21.8045
Allos. site4	21.2119	0.1409	2.5454 × 10^6^	20.7715	21.2161	21.6625
Allos. site5	21.1757	0.1174	2.5411 × 10^6^	20.7654	21.1789	21.7439
Allos. site6	21.2353	0.1524	2.5482 × 10^6^	20.7983	21.2214	21.9438
Allos. site7	21.5352	0.1706	2.5842 × 10^6^	21.0101	21.5438	22.0738
Allos. site8	21.3830	0.1535	2.5660 × 10^6^	20.9068	21.3716	21.9775

## Data Availability

The data presented in this study are available on request from the corresponding author.

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
