# Peer review of "Allostery Inhibition of BACE1 by Psychotic and Meroterpenoid Drugs in Alzheimer’s Disease Therapy"

_molecules, 2022, doi:10.3390/molecules27144372_

Round 1

Reviewer 1 Report

Authors have performed a molecular docking along with molecular dynamics to explore selected compounds based on literature against allosteric site of BACE-1, important target for AD. However, there are number of issues (mentioned below) which should be resolved before consideration of this manuscript for the publication.

  1. Title need revision. The class compounds should be reflected.
  2. Abstract need revision. Author should focus more on material and method and results.
  3. Introduction is too long which can be shortened.
  4. Author should represent results of docking in a tabular form for better understanding of readers. The column of tables can include: name of compound, class, doc score, number of hydrogen bonds, name of amino acids involved for hydrophobic, hydrogen, pi-pi interaction, etc.
  5. Author should compare the doc score of internal ligands (if present) with tested compounds.
  6. How allosteric sites were identified?
  7. The simulation time for molecular dynamics studies is too small.
  8. What is basis of selection of compounds? How many compounds are selected for molecular dynamics based on molecular docking results?
  9. Author should key findings with literature.
  10. All used software’s are freely available?
  11. It could be better if author can predict ADME and toxicity profile of selected compounds. Material and method section should be presented clearly along with suitable references. For molecular docking author could check recently published articles: Bhosale, S., & Kumar, A. (2021). Screening of phytoconstituents of Andrographis paniculata against various targets of Japanese encephalitis virus: An in-silico and in-vitro target-based approach. Current Research in Pharmacology and Drug Discovery, 2, 100043. https://doi.org/10.1016/j.crphar.2021.100043. Bhimaneni, S. P., Bhati, V., Bhosale, S., & Kumar, A. (2021). Investigates interaction between abscisic acid and bovine serum albumin using various spectroscopic and in-silico techniques. Journal of Molecular Structure, 1224, 129018. 
  12. For molecular dynamics, author could check following paper:  Kant, K., Rawat, R., Bhati, V., Bhosale, S., Sharma, D., Banerjee, S., & Kumar, A. (2021). Computational identification of natural product leads that inhibit mast cell chymase: an exclusive plausible treatment for Japanese encephalitis. Journal of Biomolecular Structure and Dynamics39(4), 1203-1212.

Author Response

Response to Reviewer 1 Comments

Point 1: Title need revision. The class compounds should be reflected.

Response 1: Allostery Inhibition of BACE1 by Psychotic and Meroterpenoid drugs in Alzheimer's Disease Therapy

Point 2: Abstract need revision. Author should focus more on material and method and results.

Response 2: : In over a century of its discovery, Alzheimer's disease (AD) has continued to be a global health concern due to its incurable nature and overwhelming increase among the older people. Lately, we gave an overview of the efforts of researchers towards identifying potent BACE1 exosite-binding antibodies and allosteric inhibitors. Herein, we apply computer-aided drug design (CADD) methods to unravel the interactions of some proposed psychotic and meroterpenoid BACE1 allosteric site inhibitors. This study is aimed at validating the allosteric potentials of these selected compounds targeted at BACE1 inhibition. Molecular docking, molecular dynamic (MD) simulations and post-MD analyses are carried out on these selected compounds which have been experimentally proven to exhibit allosteric inhibition on BACE1. The SwissDock software enabled us to identify more than five druggable pockets on the BACE1 structural surface using docking. Besides the active site region, a melatonin derivative (compound 1) previously proposed as a BACE1 allostery inhibitor showed appreciable stability at eight different subsites on BACE1. Refinement with molecular dynamic (MD) simulations shows that the identified non-catalytic sites are potential allostery sites for compound 1. The allostery and binding mechanism of the selected potent inhibitors show that the smaller the molecule, the easier the attachment to several enzyme regions. This finding hereby establishes that most of these selected compounds failed to exhibit strong allosteric binding with BACE1 with the exception of compound 1. We hereby suggest that further studies and additional identification/validation of other BACE1 allosteric compounds be done. Furthermore, this additional allosteric site investigation will help in reducing the associated challenges with designing BACE1 inhibitors while exploring the opportunities in the design of allosteric BACE1 inhibitors.  

Point 3: Introduction is too long which can be shortened.

Response 3: The introduction has been shortened with lines 55-61 of paragraph three removed.

Point 4: Author should represent results of docking in a tabular form for better understanding of readers. The column of tables can include: name of compound, class, doc score, number of hydrogen bonds, name of amino acids involved for hydrophobic, hydrogen, pi-pi interaction, etc.

Response 4: Table 2 represents predominant interactions of the some docked compounds including their binding energies as could be seen in page 10. The interactions of compound 1 are extensively discussed in this manuscript and shown in figure 6 of page 9.

Compound

Class

Binding energy in kcal/mol

Residues forming hydrogen bond

Residues forming Electrostatic interaction

Sargahydroquinoic acid

Meroterpenoids

-8.64

GLN 73, GLY 34

GLY 74, GLY73

Gamma-linolenic acid

Meroterpenoids

-7.97

GLY 34, THR 33

VAL 31, ASH 32, THR 33, GLY 34, SER 35

Ly2811376

Antipsychotic drug

-7.80

ASH 32

LEU 30, VAL 31, ASH 32, GLY 34, THR 72, GLN 73, GLY 74, LYS 107, PHE 108, LEU 119, ASP 228

Anisoperidone

Psychotic drug

-7.91

SER 35

ASH 32, SER 35, GLY 230

Point 5: Author should compare the doc score of internal ligands (if present) with tested compounds.

Response 5: The docked results were compared with the internal inhibitor, the selected drugs showed lower docking scores and binding energies relative to original drug Am6494 with the PDB ID: 6zp4 (Table 1). This is expected because the internal ligand binds strongly at the active site. This has been verified and validated computationally by our group in our two previous studies on BACE1. See references 1 and 20.

Point 6: How allosteric sites were identified?

Response 6: The methods used in the identification of the allosteric sites was mentioned in our previous reviewed work as could be seen in lines 108-113 (page 4) of this manuscript and referenced in 20 and 28. Most of these compounds were identified experimentally hence the aim of this study to computationally validate the allosteric potentials of the chosen compounds.

Point 7: The simulation time for molecular dynamics studies is too small.

Response 7: The simulation time of 120ns enabled us to adequately study the system. Figure 7 which is the RMSD plot shows that the system stabilized and converged between 80-100 ns of the simulation runs.

Point 8: What is basis of selection of compounds? How many compounds are selected for molecular dynamics based on molecular docking results?

Response 8: This was already explained in lines 131-136 of page 4. Docking analysis revealed that these compounds slightly bind at allosteric sites but are more stable at the active site. “Only compound 1, a melatonin derivative showed appreciable stability at six different allostery pockets. To evaluate how stable these subsites are, we apply MD simulations to explore the allostery dynamics and interaction of compound 1 at all the binding sites. Compound 1 binds favorably at BACE1 active region compared to the other six predicted sites”.

Point 9: Author should key findings with literature.

Response 9: Literatures are cited in page 4 lines 134, 135 (references 20, 32) and in page 7 lines 249-251 (references 33, 34, 72 ,73).

Point 10: All used software’s are freely available?

Response 10: All the software are free except Amber.

Point 11: For molecular dynamics, author could check following paper:  Kant, K., Rawat, R., Bhati, V., Bhosale, S., Sharma, D., Banerjee, S., & Kumar, A. (2021). Computational identification of natural product leads that inhibit mast cell chymase: an exclusive plausible treatment for Japanese encephalitis. Journal of Biomolecular Structure and Dynamics39(4), 1203-1212.

Response 11: The paper has been cited (page 6 line 183) and added to the reference list reference 60.

Reviewer 2 Report

The manuscript applies CADD methods to unravel the interactions of five proposed BACE1 allosteric site inhibitors. Although it is a valuable work having an interesting idea it needs some adjustment that I list below.

DETAILED REVIEW

The weaknesses of the reviewed work concern the following issues:

  • some major point:
  • The organization of work is chaotic and requires ordering. Too long introduction, presentation of results and discussions in the form of one chapter. Additionally, the conclusions are loosely related to the conducted research. An insightful reader will interpret them as prospects for the future, and not as actual conclusions from the research presented in the paper.
  • As recommended by the journal, please shorten the introduction and present the Results and Discussion section in the form of two separate chapters. Many issues presented in the introduction should be included in the discussion of the obtained results. If the authors identify a new allosteric site, this should be noted in the conclusions. This chapter can be called results and future prospects. I am convinced that such an organized presentation of the presented research aspects will make the work more transparent and will allow the reader to better understand the issues raised.
  • I do not understand any rational explanation of the selected structures set (Figure 4) for the presented study. Additionally (as shown in Figure 4) the selected compounds are structurally very different. As it is commonly known, structural differences result in different research findings. This issue requires a more detailed explanation. I suggest calculating the structure similarity coefficient (e.g. the Tanimoto coefficient using ChemMine Tools, https://chemminetools.ucr.edu/) and commenting on this research aspect in more detail.
  • some minor point:
  • Figure 4 needs to be extended by IC50 values for Anisopernidone and compound 1
  • please correct table 1 caption

I recommend publication after major revision.

Author Response

Response to Reviewer 2 Comments

The manuscript applies CADD methods to unravel the interactions of five proposed BACE1 allosteric site inhibitors. Although it is a valuable work having an interesting idea it needs some adjustment that I list below.

DETAILED REVIEW

The weaknesses of the reviewed work concern the following issues:

  • some major point:

Point 1: The organization of work is chaotic and requires ordering. Too long introduction, presentation of results and discussions in the form of one chapter. Additionally, the conclusions are loosely related to the conducted research. An insightful reader will interpret them as prospects for the future, and not as actual conclusions from the research presented in the paper.

Response 1: The introduction has been reduced with lines 55-61 of paragraph three removed. The conclusion has be improved to reflect the actual research presented in the study.

Point 2: As recommended by the journal, please shorten the introduction and present the Results and Discussion section in the form of two separate chapters. Many issues presented in the introduction should be included in the discussion of the obtained results. If the authors identify a new allosteric site, this should be noted in the conclusions. This chapter can be called results and future prospects. I am convinced that such an organized presentation of the presented research aspects will make the work more transparent and will allow the reader to better understand the issues raised.

Response 2: The conclusion has been carefully and articulately rewritten to reflect the findings of the present study. “Allosteric inhibition is a type of competitive inhibition occurring at another protein site different from the active site. In this mechanism, the inhibitor binds to a site(s) other than the active site thereby, rendering the active site unfit for the substrate to bind. In allosteric inhibition, it is a case in which moiety gets to the enzyme or protein first that blocks the other from binding. Sequel to our previous review study on  some identified allosteric inhibitors and exosite-binding antibodies of BACE1 over the last eight years (2013-2020), we hereby in this study applied CADD methods to further validate the claims as contained in the reviewed study. We selected six compounds which were reported to bind on sites different from the BACE1 active site. Some of these compounds are classified as meroterpenoids (Sargahydroquinoic acid and Gamma-Linolenic acid) and psychotic drug (Anisoperidone). Good knowledge of the inhibiting potentials and properties of BACE1 inhibitors also serves as a prerequisite for the development of molecules with high selectivity and specificity. We show in this study that besides the active site, BACE1 targeting at other subsites is plausible towards AD therapy. From the results of the molecular docking, molecular dynamic (MD) simulations and subsequent post-MD analyses, the identified regions correspond with some predicted subsites, and the small molecules (like compound 1) bind through several other BACE1 residues. The allostery and binding mechanism of the selected potent inhibitors show that the smaller the molecule, the easier the attachment to several enzyme regions. This finding hereby establishes that most of these selected compounds failed to exhibit strong allosteric binding with BACE1 with the exception of compound 1. We hereby suggest further studies and additional identification/validation of other BACE1 allosteric compounds be done. Furthermore, this additional allosteric site investigation will help in reducing the associated challenges with designing BACE1 inhibitors while exploring the opportunities in the design of allosteric BACE1 inhibitors. We further suggest that diverting attention from the conventional active site through a detailed computer-aided drug design approach would assist in designing a potential BACE1 inhibitor that might attain the approval stage for AD management”.

Point 3: I do not understand any rational explanation of the selected structures set (Figure 4) for the presented study. Additionally (as shown in Figure 4) the selected compounds are structurally very different. As it is commonly known, structural differences result in different research findings. This issue requires a more detailed explanation. I suggest calculating the structure similarity coefficient (e.g. the Tanimoto coefficient using ChemMine Tools, https://chemminetools.ucr.edu/) and commenting on this research aspect in more detail.

Response 3: The compounds were selected based on some experimental findings and a previous review study claiming the allosteric potentials of these different compounds. The methods used in the identification of the allosteric sites was mentioned in our previous reviewed work as could be seen in lines 108-107 (page 4) of this manuscript and referenced in 20 and 28. Most of these compounds were identified experimentally hence the aim of this study to computationally validate the allosteric potentials of the chosen compounds. Furthermore, as already explained in lines 131-136 of page 4. Docking analysis revealed that these compounds slightly bind at allosteric sites but are more stable at the active site. “Only compound 1, a melatonin derivative showed appreciable stability at six different allostery pockets. To evaluate how stable these subsites are, we apply MD simulations to explore the allostery dynamics and interaction of compound 1 at all the binding sites. Compound 1 binds favorably at BACE1 active region compared to the other six predicted sites”.

Calculating the structural similarity is well noted and would be considered in further studies.

Point 4: some minor point:

  • Figure 4 needs to be extended by IC50 values for Anisopernidone and compound 1
  • please correct table 1 caption

Response 4: The IC50 value of compound 1 has been included in the paper as could be seen in figure 4 while the table 1 caption has been corrected in lines 280-281.

I recommend publication after major revision.

Round 2

Reviewer 1 Report

Author has addressed all my comments in the revised manuscript. 

Reviewer 2 Report

Thank you to the authors for addressing my comments. I believe the revised manuscript improved in clarity.